# Quant VideoGen: Auto-Regressive Long Video Generation via 2-Bit KV-Cache Quantization

Haocheng Xi [* 1]   Shuo Yang [* 1]   Yilong Zhao [1]   Muyang Li [2]   Han Cai [3]   Xingyang Li [2]   Yujun Lin [3]
Zhuoyang Zhang [2]   Jintao Zhang [1]   Xiuyu Li [1]   Zhiying Xu [† 4]   Jun Wu [† 4]   Chenfeng Xu [5]   Ion Stoica [1]
Song Han [3 2]   Kurt Keutzer [1]

## Abstract

Despite rapid progress in auto-regressive video diffusion, we identify an emerging system–algorithm bottleneck that limits both deployability and capability: KV-cache memory. In auto-regressive video generation models, the KV-cache grows with history and rapidly dominates GPU memory (often $\geq$ 30 GB), blocking deployment on widely available hardware. More importantly, memory-bounded KV budgets force small working memory, which directly degrades long-horizon consistency in identity, layout, and motion, etc. To bridge this gap, we present Quant VideoGen (QVG), a training-free KV-cache quantization framework for auto-regressive video diffusion model. QVG exploits video's spatiotemporal redundancy through Semantic-Aware Smoothing to produce low-magnitude, quantization-friendly residuals. QVG further propose Progressive Residual Quantization, a coarse-to-fine multi-stage scheme that reduces quantization error while enabling a smooth quality–memory trade-off. Across LongCat-Video, HY-WorldPlay, and Self-Forcing, QVG establishes a new Pareto quality-memory frontier, reducing KV memory by up to $7.0\times$ with $<$ 4% end-to-end latency overhead and significantly better quality over baselines.

## 1. Introduction

Video diffusion models have progressed at a remarkable pace. Powered by bidirectional attention backbones (e.g., HunyuanVideo (Wu et al., 2025), Wan2.1/Wan2.2 (Wan et al., 2025)), today's systems can synthesize short clips with compelling photorealism and coherent motion. Yet this quality leap has not translated into long-horizon capability: from 2024 to 2026, mainstream bidirectional-attention video diffusion models have largely remained confined to 5-10 second generations in practical settings, leaving a persistent gap in deployment scenarios that demand minute-level, even hour-level continuity and interaction.

A central reason is the unfavorable *generation scaling*. Bidirectional video diffusion models perform inference in a way that: at each denoising step, tokens attend to both past and future frames. From a systems perspective, this induces a late-commit execution model: early frames cannot be safely output until the full window finishes denoising. Auto-regressive video diffusion thus marks a paradigm shift. By enforcing temporal causality, approaches such as CausVid (Yin et al., 2025) and Self-Forcing (Huang et al., 2025) turn the computation graph amortized, output frames depend only on retained history, so they can be incrementally committed and streamed as soon as they are produced. This shift has already opened up application regimes that are difficult to support with bidirectional attention, including live-streaming video generation (Feng et al., 2025), interactive content control (*e.g*, Matrix Game (He et al., 2025)), and long-horizon spatial exploration or 3D-consistent synthesis (*e.g.*, world model pipelines (Xiao et al., 2025)).

However, auto-regressive video models introduce a system-algorithm coupled problem: *KV-cache memory*. Specifically, auto-regressive inference accumulates a large KV-cache that must remain resident to avoid KV-cache recomputation. In practice, KV-cache quickly dominates GPU memory and becomes the binding resource well before raw compute saturates (Team et al., 2025). For instance, generating a 5-second 480p video by LongCat-Video (Team et al., 2025) re-

---

[1]University of California, Berkeley [2]Massachusetts Institute of Technology [3]NVIDIA [4]Amazon [5]The University of Texas at Austin. Correspondence to: Kurt Keutzer <keutzer@berkeley.edu>, Chenfeng Xu <xuchenfeng@utexas.edu>.

[†]This work does not relate to the position at Amazon.

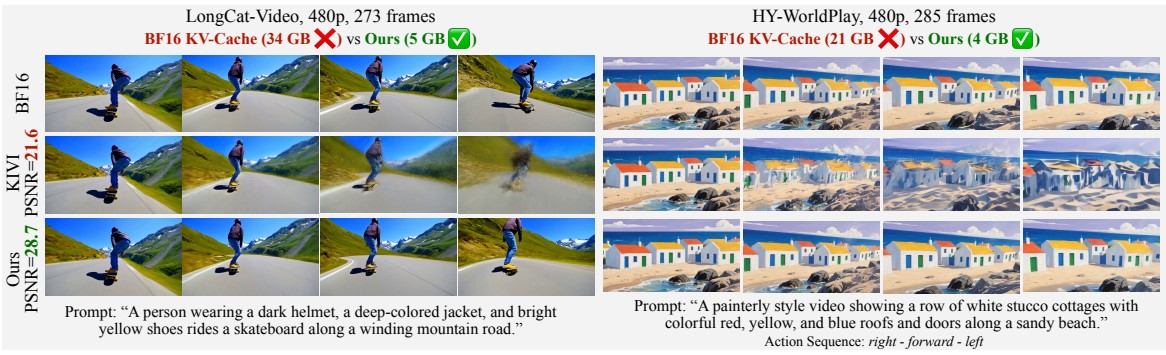

*Figure 1.* QVG makes long video generation extremely memory-efficient and maintains high video quality. On LongCat-Video and HY-WorldPlay, QVG reduces the memory footprint by up to $7\times$ and achieves a PSNR of 28.7, much better than the baseline.

quires approximately 38K tokens, corresponding to roughly 34 GB of KV-cache, which already exceeds the memory capacity of a single RTX 5090 GPU. As generation horizons lengthen, this constraint rapidly becomes hardware-limiting: even frontier world-model systems still limit generation to around 60 seconds in practice[*]. Consequently, the memory bottleneck often determines whether these models can be broadly deployed.

More importantly, KV-cache is not merely an efficiency bottleneck, it is also a capability bottleneck. We observe a strong correlation between context length and long-horizon consistency: retaining longer history in KV-cache substantially improves the preservation of identity, scene layout, and motion semantics over extended generation (Hong et al., 2025).

In this paper, we tackle the memory bottleneck by quantizing the KV-cache in auto-regressive video models. Although KV-cache quantization is mature in LLM serving with extensive work (Liu et al., 2024; Hooper et al., 2024; Kang et al., 2024; Ashkboos et al., 2024), porting these techniques naively to video diffusion leads to severe quality degradation. The gap stems from fundamentally different activation statistics: video models exhibit substantially more heterogeneous numeric distributions across both token and channel dimensions (§ 3.2), rendering the LLM-oriented assumptions brittle.

To bridge this gap, we propose **Quant VideoGen (QVG)**, a training-free KV-cache quantization framework that achieves a Pareto-frontier trade-off between generation quality and KV-cache memory footprint. The key observation behind QVG is that KV-cache of video models exhibits strong spatio-temporal redundancy (Xi et al., 2025; Yang et al., 2025), where tokens that are spatially or temporally adjacent tend to be numerically similar in latent space.

Based on this observation, we propose *Semantic-Aware Smoothing*, which groups tokens based on their similarity in latent space before quantization, to mitigate the heterogeneity in the numeric distribution. We apply the *k*-means algorithm on KV-cache along the sequence length axis to form **token groups**. By subtracting the average value of each group (i.e., the centroid), the resulting **residual tensors** have a much smaller magnitude and are more homogeneous, making it a quantization-friendly numeric distribution.

To further reduce the quantization error, we propose *Progressive Residual Quantization*, a scheme that compresses the residual tensors in multiple stages to further improve the performance. Inspired by streaming video codecs (Ma et al., 2017), which progressively encode multi-scale representations, QVG progressively groups residuals to capture information from coarse to fine granularity. This design enables a flexible trade-off between quality and compression rate by varying the number of stages.

We evaluate QVG on autoregressive video generation models, including LongCat-Video (Team et al., 2025), HY-WorldPlay (Sun et al., 2025; HunyuanWorld, 2025), and Self-Forcing (Huang et al., 2025), primarily on H100 GPUs. Across models and benchmarks, QVG consistently outperforms state-of-the-art KV-cache quantization baselines, delivering higher visual quality at lower memory cost. Concretely, QVG reduces KV-cache memory by up to $7.0\times$ while incurring minimal end-to-end latency overhead ($< 4\%$), making it practical for real-world deployment. Notably, QVG makes it possible to run HY-WorldPlay-8B on a single RTX 4090 for the first time, achieving PSNR $> 29$ relative to the BF16 reference, which was previously infeasible due to memory constraints. On the same hardware (e.g., H100), QVG further enables substantially longer effective KV-cache lengths for self-force (Huang et al., 2025), which translates into improved visual quality, even surpassing baseline BF16 under the original cache budget.

---

[*]https://labs.google/projectgenie

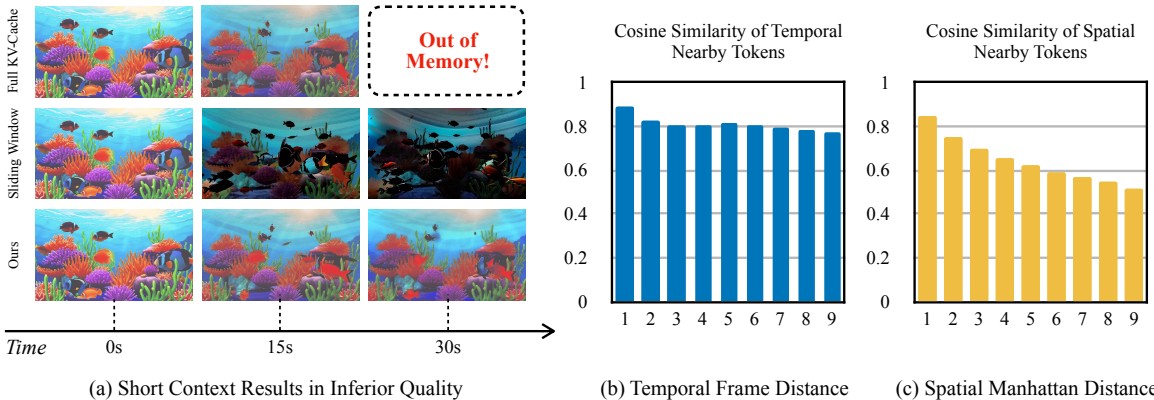

(a) Short Context Results in Inferior Quality  (b) Temporal Frame Distance  (c) Spatial Manhattan Distance

*Figure 2.* (a) Adopting full KV-cache can resolve the drifting problem but is very likely to be bottlenecked by memory. QVG can successfully generation high-quality long-videos. (b) Video diffusion models exhibit substantial spatiotemporal redundancy: tokens that are spatially or temporally adjacent have high cosine similarity, making compression feasible.

## 2. Related Work

**Auto-regressive video generation models and "memory".**
Recent video generation models are progressively shifting from bidirectional, clip-level denoising (Wan et al., 2025; Wu et al., 2025) toward chunk-level auto-regressive generation (Yin et al., 2025; Huang et al., 2025), driven by the demand for long-horizon synthesis (Bruce et al., 2024; HunyuanWorld, 2025) and real-time interaction (Feng et al., 2025). In auto-regressive video diffusion, frame chunks are generated sequentially with causal attention and KV caching, enabling substantially lower latency than offline diffusion. Beyond pushing the "faster" video generation, another major line of work pursues long-length generation. Training-free length extension methods reschedule noise or adjust temporal frequency to extrapolate pretrained models beyond their training horizon (Qiu et al., 2023; Lu et al.; Lu & Yang, 2025; Zhao et al., 2025). Complementarily, diffusion–causal hybrids (Chen et al., 2025; Song et al., 2025) improve variable-horizon conditioning and stabilize long rollouts, and are beginning to appear in streaming systems, powering applications ranging from world models and interactive agents to entertainment creation (HunyuanWorld, 2025; Sun et al., 2025; Polyak et al., 2024; Shin et al., 2025).

Some works extend long-horizon generation through explicit memory mechanisms and chunked rollouts (Henschel et al., 2025; Kodaira et al., 2025; Xiao et al., 2025; Zhang et al., 2025b; 2026). However, we emphasize that long-horizon generation is not only an algorithmic "memory" challenge, but also a system one: limited on-device hardware memory directly constrains how much algorithmic memory, *i.e.* KV-cache, can be retained. Consequently, consistent long-horizon generation is fundamentally limited by the amount of history memory we can maintain within limited hardware memory.

**Quantization-based KV-cache compression.** Compress-
ing KV-cache via quantization has been widely studied in LLMs, with diverse designs aimed at reducing memory footprint while preserving generation quality. KIVI (Liu et al., 2024) and KVQuant (Hooper et al., 2024) demonstrate that keys and values exhibit different statistics and propose a tuning-free low-bit KV-cache quantization scheme that explicitly handles heavy-tailed outliers. Beyond explicit outlier handling, rotation-based approaches such as QuaRot (Ashkboos et al., 2024) and RotateKV (Su et al., 2025) apply structured transformations to smooth activation distributions before quantization. Several works also explore token-heterogeneous strategies, e.g., prioritizing a subset of tokens to preserve quality under aggressive compression (Duanmu et al., 2024; He et al., 2024; Su & Yuan, 2025). Vector-quantization-based methods compress KV-cache by representing tokens with learned codebooks (Hooper et al., 2024; Zhang et al., 2025a; Li et al., 2025). While effective for LLMs, these methods do not explicitly exploit video-specific spatiotemporal redundancy (§ 3.3) and are not tailored to the distinct numeric characteristics that make video KV-cache quantization particularly challenging (§ 3.2).

## 3. Motivation

### 3.1. KV-Cache Bottlenecks Auto-Regressive Video Generation

**Video KV-cache is extremely memory-intensive.** In auto-regressive video diffusion models, the KV-cache grows linearly with the number of latent tokens and quickly dominates GPU memory usage for long-horizon, high-resolution generation. For a model with $L$ layers and hidden dimension $d$, storing the KV-cache for a video with spatial size $H \times W$ and temporal length $T$ requires

$$\mathrm{Mem}_{\mathrm{KV}} = 2 \cdot L \cdot (HWT) \cdot d \cdot \mathrm{Byte}_{\mathrm{BF16}}.$$

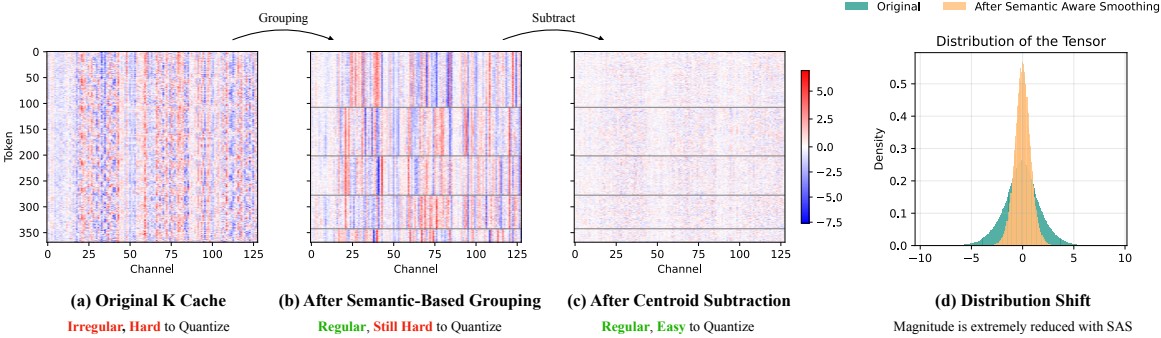

(a) Original K Cache
**Irregular**, **Hard** to Quantize

(b) After Semantic-Based Grouping
**Regular**, **Still Hard** to Quantize

(c) After Centroid Subtraction
**Regular**, **Easy** to Quantize

(d) Distribution Shift
Magnitude is extremely reduced with SAS

*Figure 3.* (a-c) Semantic-Aware Smoothing effectively smoothing the KV-cache distribution to make it more regular and quantization-friendly. We (1) group similar tokens together based on their semantic similarity and (2) subtract the centroid for each group to smooth the distribution. (d) The magnitude is significantly reduced and more concentrated around 0, making it much easier to be quantized.

For example, in LongCat-Video (Team et al., 2025), a 5-second 480p context corresponds to roughly 38K latent tokens, resulting in a KV-cache memory footprint of about 34 GB, compared to only 27 GB for model parameters. Thus, KV-cache capacity is the primary bottleneck for long-horizon video generation.

**Short-context results in inferior generation quality.** Existing auto-regressive video systems commonly enforce a fixed-length context window in their default inference configurations. For example, Wan distilled Self-Forcing (Huang et al., 2025) uses a rolling KV-cache with a default window of 21 latent frames, and HY-WorldPlay (HunyuanWorld, 2025; Sun et al., 2025) retains a compact memory of only 20 frames. This truncation is primarily driven by GPU memory concerns, where full-context KV-cache becomes quickly infeasible for long-horizon generation. However, bounding the context effectively shrinks the model's working memory, which can exacerbate long-horizon drift and limit revisitability and temporal consistency, as visualized in Figure 2(a). Therefore, we hope to address this memory bottleneck by quantizing the KV-cache to a lower bit-precision.

### 3.2. Video KV-Cache Quantization is Challenging

Quantization maps floating-point values into low-bit values to reduce the memory footprint. In this paper, we consider symmetric per-group integer quantization with bit-width $b$. The quantize and dequantize process is formulated as

$$X_{\text{INT}}, S_X = Q(X_{\text{BF16}}), \quad \hat{X} = S_X \cdot X_{\text{INT}} \quad (1)$$

where

$$X_{\text{INT}} = \left\lfloor \frac{X_{\text{BF16}}}{S_X} \right\rceil, \quad S_X = \frac{\max(|X_{\text{BF16}}|)}{2^{b-1} - 1} \quad (2)$$

For any $x \in X_{\text{BF16}}$, the quantization error satisfies

$$|x - \hat{x}| = S_X \cdot \text{RoundErr}\left(\frac{x}{S_X}\right) \leq \frac{S_X}{2}, \quad (3)$$

where $\text{RoundErr}(u) = |u - \lfloor u \rceil|$. Assuming each elements are independent, the expected error obeys

$$\mathbb{E}[|x - \hat{x}|] \propto S_X, \quad (4)$$

so large scaling factors (e.g., caused by outliers) lead to large quantization error.

Crucially, this effect is exacerbated in auto-regressive video generation, where the KV-cache exhibits highly dynamic numeric ranges across both tokens and channels. Empirically, on Wan distilled Self-Forcing and LongCat-Video, we observe that $\max|K| \sim 1e2$ and $\max|V| \sim 1e3$. Beyond the exceptionally large numerical range, we also observe irregularity across the channel dimension at the token level: channels that are outliers for some tokens may not be outliers for others, as shown in Figure 3(a-b). This behavior is intrinsic to video models: tokens correspond to diverse spatial regions and motion patterns whose relevance evolves over time, leading attention projections to produce strongly non-uniform activation scales across space and time.

### 3.3. Video KV-Cache is Highly Redundant

**Spatiotemporal redundancy in video tokens.** Video content exhibits strong redundancy across both time and space, and this redundancy is reflected in the latent tokens. As shown in Figure 2(b), for a fixed spatial location (i.e., the same patch index), tokens from adjacent frames often remain highly similar because large portions of a scene are static or evolve smoothly. Shown in Figure 2(c), spatially nearby patches also map to highly similar latent tokens: when two nearby patches have high pixel-level similarity, their corresponding latent tokens typically exhibit high cosine similarity as well.

**Progressive encoding of videos.** Videos exhibit an inherent progressive structure that naturally supports residual-based representations. Temporal coherence allows most frames to be predicted at a coarse level, capturing global layout and dominant motion, while finer details are introduced incre-

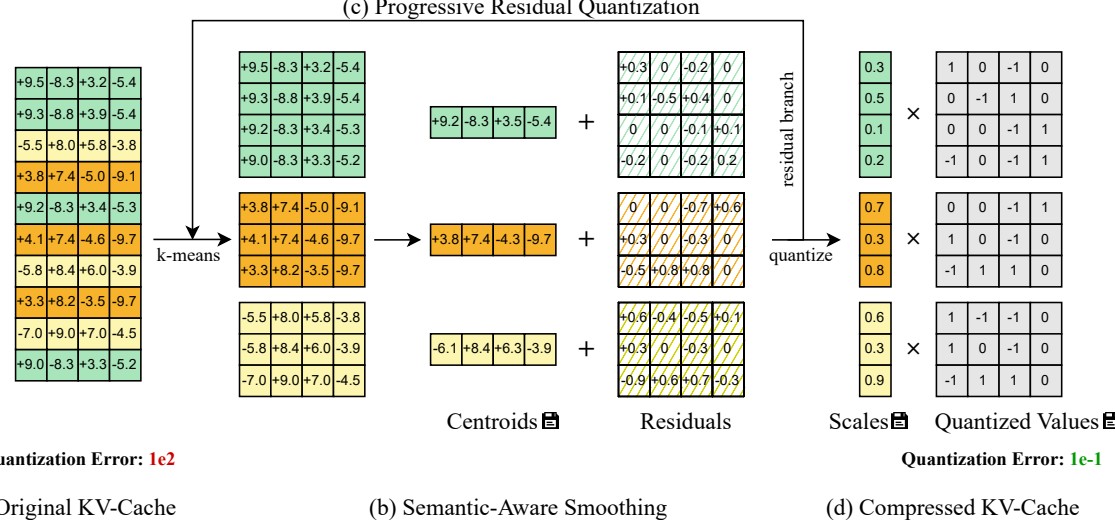

(c) Progressive Residual Quantization

(a) Original KV-Cache      (b) Semantic-Aware Smoothing      (d) Compressed KV-Cache

*Figure 4.* Overview of QVG framework. (a) Original tensor's distribution is irregular and hard to quantize. (b) Semantic-Aware Smoothing groups similar tokens and subtracts centroids for each group to make the residual quantization friendly. (c) Progressive Residual Quantization further lowers quantization error by iteratively applying Semantic-Aware Smoothing algorithm. (d) The final residual tensor becomes much easier to quantize and has a much lower quantization error.

mentally through residuals. As a result, video content can be encoded progressively from coarse scene structure and color composition to fine-grained textures and high-frequency details. We quantitatively showcase this in Table 1.

## 4. Methodology

Based on these insights, we introduce QVG as a reliable KV-cache quantization technique for video generation. We first introduce Semantic-Aware Smoothing in § 4.1 to smooth the KV-cache distribution and make it more quantization-friendly, as visualized in Figure 3. We then introduce Progressive Residual Quantization in § 4.2 to further improve the generation quality. Besides, we also discuss several algorithm-system co-design optimizations in § 4.3. In Figure 4 we provide an overview of our proposed method.

### 4.1. Semantic-Aware Smoothing

As discussed in § 3.3, video tokens exhibit strong spatiotemporal redundancy. Semantic-Aware Smoothing exploits this redundancy to form semantically similar groups, reducing their magnitude and enabling accurate low-bit quantization.

**Semantic-based grouping.** QVG processes the KV-cache in a chunk-by-chunk manner, where each chunk consists of tokens from a few frames. Consider a chunk containing $N$ tokens (e.g., $N = HWT_c$ for $H$ height, $W$ width, and $T_c$ latent frames). Let $\mathbf{X} \in \mathbb{R}^{N \times d}$ denote the corresponding KV-cache, with $d$ being the head dimension. We partition the $N$ tokens into $C$ groups using the *k*-means algorithm based on their hidden representations. This produces a set

of disjoint groups $\mathcal{G} = \{\mathcal{G}_1, \mathcal{G}_2, \dots, \mathcal{G}_C\}$, where each group $\mathcal{G}_i$ contains tokens with similar hidden-representations. As visualized in Figure 3(a), tokens within the same group exhibit significantly more homogeneous value distributions. We represent the mean value of each group (also known as centroid) as $C_i \in \mathbb{R}^d$.

**Residual computation via centroid subtraction.** Then for each group $\mathcal{G}_i$, to make the distribution smoother, we subtract the centroid and obtain the residual:

$$\boldsymbol{R}_i = \boldsymbol{X}_{\mathcal{G}_i} - C_i, \quad \boldsymbol{R}_i \in \mathbb{R}^{|\mathcal{G}_i| \times d}, \tag{5}$$

where $\boldsymbol{X}_{\mathcal{G}_i}$ refers to the matrix formed by tokens $\in \mathcal{G}_i$. As discussed in § 3.2, quantization error is proportional to the maximum value in the group. Due to the *k*-means clustering, these large values are expected to be shared across the group and are captured by $C_i$. Therefore, as shown in Figure 3, by subtracting the centroid the maximum value in each group becomes much smaller, which results in lower quantization error. As visualized in Figure 6, we successfully reduce the quantization error of Key Cache by $\sim 6.9\times$, and reduce the quantization error of Value Cache by $\sim 2.6\times$ on all precision choices. This proves the effectiveness of our method.

**Summarization and visualization.** Semantic-Aware Smoothing process can be represented as follows:

$$\boldsymbol{R}, \mathbf{C}, \boldsymbol{\pi} = \text{SA-Smoothing}(\boldsymbol{X}, C), \tag{6}$$

where $\boldsymbol{R} \in \mathbb{R}^{N \times d}$ is the residual tensor, $\mathbf{C} \in \mathbb{R}^{C \times d}$ is the centroids, and $\boldsymbol{\pi} \in \{1, \dots, C\}^N$ is the assignment vector that denotes the centroid assignment of each token.

*Table 1.* Quality and Compression results of QVG and baselines. QVG-Pro achieves $4.97\times \sim 5.20\times$ compression ratio, while achieving much better fidelity scores than all baselines. QVG further pushes the compression ratio to $6.94\times \sim 7.05\times$ and still maintains near-lossless video quality scores.

| Method | Compression Ratio (BF16) | PSNR | SSIM | LPIPS | Background Consistency | Image Quality | Subject Consistency | Aesthetic Quality |
|---|---|---|---|---|---|---|---|---|
| *LongCat-Video-13B* | | *INT2 KV Cache, 480p* | | | 96.22 | 72.72 | 95.51 | 64.83 |
| RTN | 6.40× | 20.872 | 0.719 | 0.203 | 84.84 | 59.60 | 70.63 | 43.38 |
| KIVI | 6.40× | 20.317 | 0.719 | 0.208 | 84.84 | 38.10 | 75.25 | 41.58 |
| Quarot | 6.40× | 21.573 | 0.759 | 0.171 | 86.12 | 50.70 | 80.61 | 49.49 |
| **QVG-Pro** | **4.97×** | **30.376** | **0.935** | **0.048** | 96.20 | 71.74 | 94.92 | 63.88 |
| **QVG** | **6.94×** | 28.716 | 0.909 | 0.065 | 95.06 | 71.47 | 94.11 | 62.22 |
| *LongCat-Video-13B* | | *INT4 KV Cache, 480p* | | | 96.22 | 72.72 | 95.51 | 64.83 |
| RTN | 3.55× | 32.984 | 0.940 | 0.045 | 96.13 | 72.14 | 95.53 | 64.57 |
| KIVI | 3.55× | 32.158 | 0.946 | 0.040 | 96.16 | 72.67 | 95.26 | 64.53 |
| Quarot | 3.55× | 33.744 | 0.960 | 0.033 | 95.48 | 72.47 | 95.48 | 64.92 |
| **QVG-Pro** | **3.05×** | **37.095** | **0.977** | **0.024** | 96.67 | 72.66 | 95.44 | 64.93 |
| **QVG** | **3.72×** | 37.141 | 0.978 | 0.024 | 95.94 | 72.34 | 94.34 | 64.88 |
| *HY-WorldPlay-8B* | | *INT2 KV Cache, 480p* | | | 97.92 | 74.33 | 97.90 | 69.85 |
| RTN | 6.40× | 24.199 | 0.696 | 0.229 | 96.16 | 71.86 | 96.08 | 69.15 |
| KIVI | 6.40× | 24.272 | 0.701 | 0.230 | 96.95 | 71.40 | 95.89 | 68.19 |
| Quarot | 6.40× | 25.207 | 0.738 | 0.205 | 97.34 | 72.26 | 96.64 | 69.38 |
| **QVG-Pro** | **5.20×** | **31.562** | **0.923** | **0.069** | 98.00 | 74.15 | 97.96 | 69.45 |
| **QVG** | **7.05×** | 29.174 | 0.882 | 0.094 | 97.98 | 73.87 | 97.90 | 69.80 |
| *HY-WorldPlay-8B* | | *INT4 KV Cache, 480p* | | | 97.92 | 74.33 | 97.90 | 69.85 |
| RTN | 3.55× | 33.634 | 0.948 | 0.056 | 97.98 | 74.26 | 97.87 | 70.13 |
| KIVI | 3.55× | 33.768 | 0.950 | 0.055 | 97.95 | 74.30 | 97.95 | 69.92 |
| Quarot | 3.55× | 33.997 | 0.951 | 0.053 | 97.97 | 74.33 | 97.90 | 69.76 |
| **QVG-Pro** | **3.15×** | **35.109** | **0.960** | **0.048** | 97.93 | 74.30 | 97.88 | 69.45 |
| **QVG** | **3.75×** | 34.454 | 0.954 | 0.051 | 97.96 | 74.23 | 97.90 | 69.66 |

In Figure 3, we provide visualizations to illustrate the effectiveness of Semantic-Aware Smoothing. We first directly visualize the tensor in a 2D plot in Figure 3 (a)-(c), and use gray lines to separate different groups for better visualization. As indicated in Figure 3 (b), the distribution becomes more regular after the semantic-based grouping. And as indicated in Figure 3 (c) and (d), the magnitude becomes much smaller after applying the Semantic-Aware Smoothing algorithm, confirming that Semantic-Aware Smoothing markedly compresses the dynamic range and is well-suited for low-bit quantization.

### 4.2. Progressive Residual Quantization

Motivated by the progressive structure of videos (§ 3.3), we then design Progressive Residual Quantization to further push the quantization error down in a coarse-to-fine manner. Given the output of Semantic-Aware Smoothing, Progressive Residual Quantization iteratively re-quantize the residual tensor to capture finer-grained details.

**Progressive residual refinement.** Formally, let $\boldsymbol{R}^{(0)} = \boldsymbol{X}$ denote the initial input, and let $T$ be the total number of stages, each using the same number of groups $C$. At each

stage $t$, we apply semantic-aware smoothing on the residual tensor to get a new one:

$$\boldsymbol{R}^{(t)}, \mathbf{C}^{(t)}, \boldsymbol{\pi}^{(t)} = \text{SA-Smoothing}(\boldsymbol{R}^{(t-1)}, C).$$

After $T$ stages, we obtain the final residual tensor $\boldsymbol{R}^{(T)}$. We quantize the final output into low-bit representation:

$$\boldsymbol{X}_{\text{INT}}, S_{\boldsymbol{X}} = Q(\boldsymbol{R}^{(T)})$$

and store it into global memory. We also store all $\mathbf{C}^{(t)}$ and $\boldsymbol{\pi}^{(t)}$ with $1 \leq t \leq T$, while all residuals $\boldsymbol{R}^{(t)}$ are treated as intermediate results and discarded.

Each stage focuses on re-quantizing the remaining residuals, enabling Progressive Residual Quantization to progressively model information from coarse semantic structures to fine-grained variations, thereby leading to lower quantization error. The whole pipeline is visualized in Figure 4.

**The dequantization process.** We first describe reconstruction for Semantic-Aware Smoothing, which naturally extends to the Progressive Residual Quantization setting.

Given the stored centroids $\mathbf{C}$ and assignment vector $\boldsymbol{\pi}$, each token is reconstructed by adding back its assigned centroid

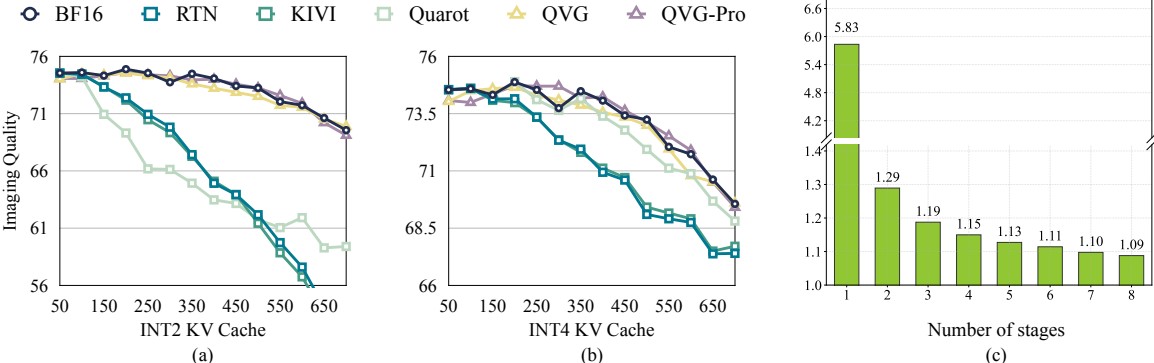

*Figure 5.* (a–b) Imaging Quality over long-horizon generation on Self-Forcing Model. Both QVG and QVG-Pro preserve near-lossless quality, while prior baselines degrade drastically. (c) The first stage of Progressive Residual Quantization yields the most significant reduction in MSE. Subsequent stages further reduce the error, but with diminishing returns.

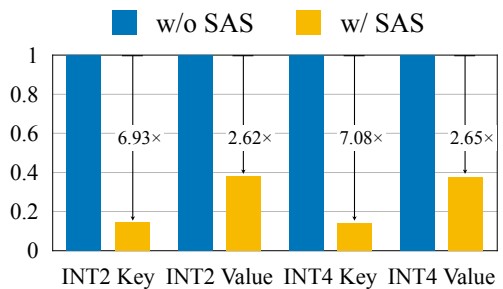

*Figure 6.* Semantic-Aware Smoothing effectively reduces the quantization error by $\sim 6.9\times$ and $\sim 2.6\times$ for keys and values, respectively. Keys has a higher MSE reduction since values cache are more irregular than keys cache.

to the corresponding residual. Specifically, let $\boldsymbol{R}$ denote the residual tensor, the reconstruction of a input token at index $1 \leq i \leq N$ is given by adding back the assigned centroid to the corresponding residual:

$$\hat{\boldsymbol{X}}_{\mathcal{G}_i} = R_i + \mathbf{C}_{\boldsymbol{\pi}_i}. \tag{7}$$

For Progressive Residual Quantization, reconstruction proceeds by iteratively applying this operation from stage $T$ to stage $1$. Starting from the quantized output $\boldsymbol{X}_{\text{INT}}$ and $S_{\boldsymbol{X}}$, we first dequantize then progressively restore $\hat{\boldsymbol{X}}^{(T-1)}, \ldots, \hat{\boldsymbol{X}}^{(0)}$, where $\hat{\boldsymbol{X}}^{(0)}$ corresponds to the final reconstructed tensor. This process is exactly the replay of the progressive quantization method.

### 4.3. Efficient Algorithm-System Co-design

**Fast $k$-means with streaming centroid caching.** While $k$-means clustering is essential to semantic-aware smoothing, its iterative procedure and the $k$-means++ initialization can introduce non-trivial latency in streaming inference. We propose a centroid caching approach to accelerate by initializing the centroid of a new video chunk using the assignment

strategy of the previous chunk. This strategy reduces the $k$-means overhead by $3\times$.

**Dequantization kernel.** We implement a fused kernel that dequantizes the tensor and adds back the assigned centroids for all stages in Progressive Residual Quantization. The intermediate result is stored in registers to avoid repeatedly reading it from global memory.

## 5. Experiments

### 5.1. Setup

**Models.** We evaluate QVG on open-sourced auto-regressive video generation models including LongCat-Video-13B (Team et al., 2025), HY-WorldPlay-8B (HunyuanWorld, 2025), and Self-Forcing-Wan-1.3B (Huang et al., 2025) to generate videos with 480p resolution. LongCat-Video-13B conducts a video continuation task based on a fixed length context window of 73 frames and repeatedly generates the next 20 frames. HY-WorldPlay-8B and Self-Forcing-Wan-1.3B condition on the entire history of previously generated frames and generate new video segments in a chunk-wise manner, with chunk sizes of 12 and 16 frames, respectively.

**Metrics.** We evaluate the fidelity compared with the BF16 KV-cache baseline using PSNR, LPIPS, and SSIM. We evaluate the perceptual quality of the generated videos using VBench (Huang et al., 2023), and report the Background Consistency, Image Quality, Subject Consistency, and Aesthetic Quality. We report the KV-cache compression ratio to measure the memory footprint reduction. We also report the incurred overhead in the end-to-end generation pipeline. For the similarity experiments on LongCat-Video-13B, we report the number of the first generated chunk as the content starts to diverge while maintaining the same quality. All other metrics are reported under the long-generation setting.

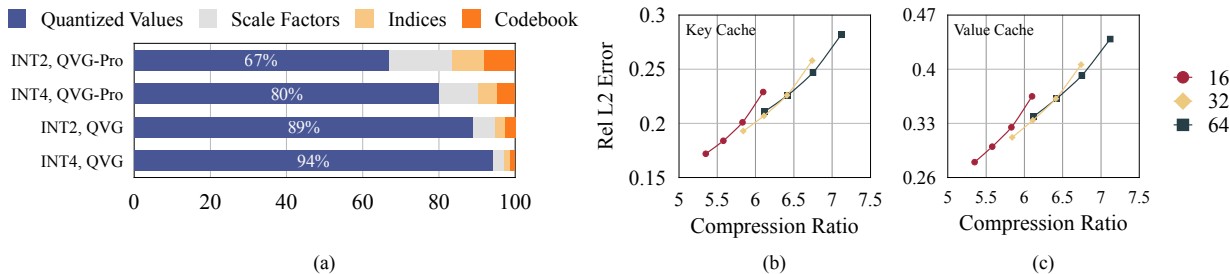

*Figure 7.* (a) Memory usage decomposition of QVG. (b-c) Trade-off curve of quantization block size for KV Cache.

**Datasets.** We use the prompt suite provided by the MovieGen benchmark (Polyak et al., 2024). Specifically, we follow Self-Forcing's official prompt settings[†].

**Baselines.** We compare QVG with Round-to-Nearest Quantization (RTN), KIVI (Liu et al., 2024), and QuaRot (Ashkboos et al., 2024) as our baselines. For QuaRot, we only implement its KV cache quantization part and do not quantize the weights and activations. We use block size 16 settings for fair comparison.

**Implementation.** We implement QVG with customized CUDA and Triton kernels and benchmark on NVIDIA H100 GPUs (CUDA 12.8). We use streaming chunk-wise compression to quantize KV-cache once per chunk and avoid re-compression drift, and adopt pre-RoPE key caching for more quantization-friendly key distributions (Hooper et al., 2024); we adopt FP8 E4M3 per-group scaling factors to reduce overhead. We evaluate INT2/INT4 under two configurations: QVG using $S=1$ and $B=64$, and QVG-Pro using $S=4$ stages and group size $B=16$. We set number of centroids $K=256$ to store assignment vectors in `uint8`.

## 5.2. Quality Evaluation

In this section, we report the results of QVG on LongCat-Video-13B, HY-WorldPlay-8B, and Self-Forcing-Wan.

As shown in Table 1, we report the results on LongCat-Video-13B and HY-WorldPlay-8B. QVG-Pro consistently achieves the best fidelity scores, while QVG delivers the largest compression ratios with only marginal quality degradation. On all VBench metrics, both QVG and QVG-Pro achieves a near-lossless performance, while all baselines exhibit huge degradation, especially under the INT2 quantization setting. Notably, our method achieves 28.716 PSNR under $6.94\times$ compression ratio for LongCat-Video-13B, and 29.174 PSNR under $7.05\times$ compression ratio for HY-WorldPlay-8B. These results demonstrate that our method can generate substantially higher-quality long videos with improved memory efficiency.

---

[†]https://github.com/guandeh17/Self-Forcing/blob/main/prompts/MovieGenVideoBench_extended.txt

We report the performance of the Self-Forcing model in Figure 5(a). Specifically, we measure the Image Quality score every 50 frames along long video sequences to evaluate whether QVG can mitigate long-horizon drift. While the BF16 KV-cache baseline also exhibits moderate quality degradation, both Quant VideoGen and Quant VideoGen-Pro maintain near-lossless quality even when extending to 700 frames. In contrast, all other baselines experience a sharp degradation after approximately 100 frames. These results demonstrate the effectiveness of our method in resisting long-horizon drift.

## 5.3. Efficiency Evaluation

**Memory usage breakdown.** We analyze the memory footprint of QVG in detail by decomposing it into four components: quantized values, assignment vector, centroids, and scaling factors. As shown in Figure 7(a), under both INT2 and INT4 precision, quantized values account for the majority of memory usage ($\geq 65\%$). Notably, QVG allocates a larger proportion to quantized values compared to QVG-Pro, which results in a higher compression ratio.

**End-to-end latency.** We evaluate the end-to-end latency to quantify the overhead introduced by quantization and dequantization in our method. On LongCat, QVG increases the overall generation time by 2.1%. On HY-World, the end-to-end overhead is 1.5%, while on Self-Forcing it is 4.3%. These results indicate that QVG introduces only modest latency overhead and does not slow down the overall generation pipeline.

## 5.4. Sensitivity Test

**Number of quantization stages.** We study the impact of the number of stages in Progressive Residual Quantization on the reduction ratio of MSE. As shown in Figure 5(b-c) the first stage provides the dominant reduction in error, resulting in $5.83\times$ MSE reduction compared with naive quantization method. Although subsequent stages decrease MSE by at least $1.10\times$, their contributions gradually decreases as the stage count grows.

**Quantization group size.** We test the impact of quantiza-

tion block size on the performance of QVG, ranging from 16 to 64. We vary the number of kmeans stages from 1 to 4 to get a trade-off curve. A larger block size leads to a higher compression ratio, but also lower quality. As shown in Figure 7(b-c), a block size of 64 achieves the best trade-off, while a block size of 16 guarantees the best quality.

## 6. Conclusion

In this paper, we propose Quant VideoGen (QVG), a training-free KV-cache quantization framework that leverages video-specific spatiotemporal redundancy to mitigate the memory bottleneck in auto-regressive video generation. We propose Semantic-Aware Smoothing that groups semantically similar tokens and subtracts group centroids to produce quantization-friendly residuals. We then propose Progressive Residual Quantization to further reduce quantization error in a coarse-to-fine manner. Across multiple video models and benchmarks, QVG achieves up to $7.04\times$ KV-cache compression with $4\%$ latency overhead while preserving near-lossless visual quality. These results demonstrate that QVG enables practical, memory-efficient long-video and world generation.

## Impact Statement

This paper presents work that aims to advance the field of Machine Learning. There are many potential societal consequences of our work, none of which we feel must be specifically highlighted here.

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

## A. Long Video and High-Resolution Evaluation

### A.1. Long Video Generation

To further verify that QVG preserves long-horizon quality at scale, we extend the Self-Forcing evaluation to 1400 frames (about 90 seconds) under the same generation setup as in the main paper. We report the VBench Image Quality of BF16, KIVI, QuaRot, and QVG under the INT2 quantization setting in Table 2. As shown, QVG consistently tracks the BF16 reference as the video length grows, while KIVI and QuaRot degrade rapidly: at 1400 frames, KIVI drops from 67.57 (at 350 frames) to 35.85, whereas QVG maintains on-par with BF16. These results confirm that QVG effectively mitigates long-horizon drift and remains robust well beyond the lengths reported in the main paper.

*Table 2.* VBench Image Quality (%) at increasing video lengths.

| Method | 350 frames | 700 frames | 1050 frames | 1400 frames |
|--------|-----------|-----------|------------|------------|
| BF16 | 74.33 | 70.56 | 68.80 | 65.87 |
| KIVI | 67.57 | 57.18 | 44.31 | 35.85 |
| QuaRot | 48.33 | 45.17 | 45.58 | 44.45 |
| QVG | **74.36** | **69.52** | **67.23** | **67.28** |

### A.2. Scalability to Higher Resolutions

The main paper evaluates QVG at 480p resolution. To assess scalability beyond 480p, we evaluate QVG on LongCat-Video at 720p while keeping the same centroid configuration as in 480p. As shown in Table 3, QVG consistently outperforms the baseline quantization methods and maintains the same error bounds. Notably, we do not need to increase the number of centroids when moving from 480p to 720p. The reason is that QVG performs quantization on fixed-size KV-cache chunks, rather than on all tokens of a frame: when the spatial resolution increases, each frame contributes more tokens, so the number of frames per chunk decreases accordingly, while the per-chunk clustering scale remains unchanged.

*Table 3.* Quality comparison on LongCat-Video at 720p under INT2 quantization.

| Method | PSNR ↑ | SSIM ↑ | LPIPS ↓ |
|--------|--------|--------|---------|
| KIVI | 17.66 | 0.5518 | 0.3441 |
| QuaRot | 20.88 | 0.6659 | 0.2395 |
| QVG | **25.99** | **0.8174** | **0.1177** |

## B. Additional Hyperparameter Ablations

### B.1. Sensitivity to Chunk Size

We profile the runtime overhead of the *k*-means clustering and quantization pipeline of QVG under different chunk sizes on Self-Forcing with INT2 quantization. As shown in Table 4, the relative overhead remains small across a wide range of chunk sizes, peaking at only 3.3% of the end-to-end runtime. Larger chunk sizes are clearly more favorable for memory efficiency, yielding both lower overhead and higher compression ratio. This is because clustering and metadata costs are better amortized over more tokens within each chunk. Overall, the results suggest that QVG is practical across different chunk sizes, while larger chunks lead to better compression ratio.

*Table 4.* Effect of chunk size on overhead and compression ratio.

| Chunk Size | Frames per Chunk | Per-Layer KV Memory | Compression Ratio | Overhead |
|-----------|------------------|---------------------|-------------------|----------|
| 37440 | 24 | 220 MB | 7.0× | 1.3% |
| 18720 | 12 | 230 MB | 6.7× | 2.0% |
| 9360 | 6 | 264 MB | 5.8× | 3.3% |

## B.2. Sensitivity to Codebook Size $K$

The choice of $K$ should balance clustering quality and metadata overhead. Increasing $K$ improves clustering quality, but also increases the index/codebook overhead and reduces the compression ratio. In Table 5, we sweep over the codebook size $K \in \{64, 128, 256, 512\}$ while fixing the number of Progressive Residual Quantization stages to 1, and report results under both INT4 and INT2 settings. We report the resulting compression ratio together with the reconstruction-error reduction over naive quantization for the key and value caches, denoted as K-Imp and V-Imp; larger values indicate lower MSE and better reconstruction.

We observe that both $K=128$ and $K=256$ provide good trade-offs. We choose $K=256$ in the final design because it allows the assignment vector to be stored naturally in `uint8`, i.e., each token's cluster ID fits in one byte.

*Table 5.* Effect of codebook size $K$ under INT4 (left) and INT2 (right) quantization, with single-stage Progressive Residual Quantization.

| INT4 | | | | | INT2 | | | |
|---|---|---|---|---|---|---|---|---|
| $K$ | Comp. Ratio | K-Imp | V-Imp | | $K$ | Comp. Ratio | K-Imp | V-Imp |
| 512 | 3.819 | 6.558 | 2.391 | | 512 | 7.307 | 6.428 | 2.356 |
| 256 | 3.881 | 5.944 | 2.229 | | 256 | 7.539 | 5.834 | 2.199 |
| 128 | 3.917 | 5.402 | 2.083 | | 128 | 7.676 | 5.307 | 2.056 |
| 64 | 3.939 | 4.912 | 1.946 | | 64 | 7.760 | 4.832 | 1.924 |

# C. Additional Hardware and System Overhead Analysis

## C.1. Overhead on Consumer-Grade GPUs

To verify that QVG remains lightweight beyond datacenter hardware, we provide a breakdown of the end-to-end latency and the QVG-related overhead on Self-Forcing for both an NVIDIA H100 and an NVIDIA RTX 5090. As shown in Table 6, the quantization overhead remains very small on both platforms, staying below 2% of the end-to-end latency. The overhead is slightly higher on the RTX 5090 than on the H100, which is consistent with the slower consumer-class GPU making the quantization cost a slightly larger fraction of the total runtime. Overall, these results show that QVG introduces only marginal extra cost across different hardware settings.

*Table 6.* Latency and QVG overhead on H100 vs. RTX 5090.

| Platform | End-to-end | QVG extra cost | Total overhead |
|---|---|---|---|
| H100 | 43 s | 0.74 s | 1.7% |
| RTX 5090 | 72 s | 1.34 s | 1.9% |

## C.2. Scalability to Larger Batch Sizes

We additionally study how the relative overhead of QVG scales with batch size. The overhead of QVG is mainly determined by chunk size, rather than directly by resolution, sequence length, or batch size. Because QVG operates on fixed-size KV-cache chunks, increasing resolution or sequence length mainly increases the number of chunks, while the per-chunk cost stays unchanged. Similarly, as batch size increases, both the model forward and QVG's bookkeeping scale roughly linearly, while the model forward remains dominant. As a result, larger batch sizes increase absolute latency but have little effect on QVG's percentage overhead. Table 7 confirms this on Self-Forcing: the relative overhead remains stable around 1.6%–1.7% from batch size 1 to 5.

*Table 7.* Latency and QVG overhead at different batch sizes.

| Batch Size | End-to-end | QVG extra cost | Total overhead |
|---|---|---|---|
| 1 | 43 s | 0.74 s | 1.7% |
| 2 | 86 s | 1.4 s | 1.6% |
| 5 | 217 s | 3.7 s | 1.7% |

