# OpenReview forum: "Quant VideoGen: Auto-Regressive Long Video Generation via 2-Bit KV-Cache Quantization"
_ICML.cc/2026/Conference — ICML 2026 regular_

### Official Review · Reviewer_A2Vs · 2026-02-13

**Soundness:** 2
**Presentation:** 3
**Significance:** 3
**Originality:** 2
**Overall Recommendation:** 4
**Confidence:** 2

**Summary:**

This paper targets the memory bottleneck in long-horizon autoregressive video diffusion/decoding by aggressively compressing the attention KV cache at inference time. The core idea is to make extreme low-bit KV quantization (down to 2-bit) viable for video generation by first reshaping KV statistics into a quantization-friendly form. The method is training-free and relies on a semantic-aware token grouping procedure that subtracts group centroids to produce small residuals, plus a progressive residual quantization scheme that refines residual representations over multiple stages. The paper also includes practical system optimizations (e.g., caching and fused kernels) to keep latency overhead small. Experiments across several autoregressive video generation settings suggest the approach achieves a better quality–memory trade-off than prior KV quantization baselines, enabling longer effective context without severe drift.

**Compliance With Llm Reviewing Policy:**

Affirmed.

**Final Justification:**

The authors have done additional experiments to improve results and provide more concrete evidence of the method working.

**Key Questions For Authors:**

How sensitive is performance to the clustering hyperparameters (number of clusters, chunk length, initialization/caching policy), and can you provide a recommended default that transfers across models and resolutions?

What are the failure modes? Please provide qualitative and quantitative analysis on scenes with rapid motion, scene cuts, or low redundancy, where semantic grouping may be unstable.

How does the method scale in wall-clock time and memory bookkeeping at higher resolutions and longer sequences (including batch > 1), and what is the break-even point compared with simpler quantization schemes?

Can you demonstrate generalization to at least one additional autoregressive video generation architecture/tokenization setup not covered in the current experiments, to support model-agnostic claims?

What parts of the method contribute most to long-horizon stability: centroid subtraction, progressive refinement, or the systems-level fused reconstruction? A more targeted ablation on long-horizon metrics would clarify causality.

**Limitations:**

yes

**Strengths And Weaknesses:**

Strengths:

Clear practical motivation: KV-cache memory is a concrete deployment bottleneck for long-horizon video generation, and reducing it can directly translate into longer context and improved temporal consistency.

Training-free, plug-and-play design: The approach does not require retraining or model modifications, which is valuable for rapid adoption and for models where training is expensive or unavailable.

Method aligns with video statistics: The paper explicitly argues and empirically supports that video tokens exhibit strong spatiotemporal redundancy and heterogeneous activation patterns, motivating token-wise semantic grouping and residualization rather than applying LLM-style KV quantization directly.

Strong empirical quality–compression results: Compared with representative KV quantization baselines, the proposed method appears to maintain substantially higher perceptual fidelity at very low bit-widths, and the long-horizon degradation curves suggest improved stability over time.

Systems awareness: The inclusion of implementation details (e.g., fused dequantization / reconstruction kernels and caching of clustering state across chunks) indicates attention to end-to-end inference cost, not only accuracy metrics.

Weaknesses:

Conceptual novelty is moderate: The method is a thoughtful combination of known ideas (clustering / centroid subtraction, residual quantization, progressive refinement) adapted to KV-cache compression for video; the primary novelty is in the specific formulation and empirical demonstration rather than a fundamentally new principle.

Sensitivity and robustness are not fully characterized: The approach introduces several algorithmic choices (number of clusters, chunking strategy, number of progressive stages, initialization/caching heuristics). While some ablations may exist, the paper would benefit from clearer guidance on stable settings across models and resolutions, and from stress tests where clustering quality degrades (e.g., highly dynamic scenes, rapid shot changes).

Overhead and scaling concerns: Even with optimizations, clustering-based procedures may become nontrivial at higher resolutions, longer sequences, or larger batch sizes. A more detailed breakdown of time and memory overhead across regimes (and comparison against simpler heuristics) would strengthen the systems claim.

Breadth of evaluation: The experiments focus on a limited set of models and settings. It is unclear how well the approach generalizes to other tokenizers/architectures, different attention implementations, or alternative decoding pipelines.

Reproducibility details: For a systems-oriented contribution, the paper should be very explicit about implementation specifics (kernel design assumptions, GPU types, precision formats, caching policy, and exact evaluation protocol) to make reproduction and fair comparison easier.

---

> ### Author Rebuttal · Authors · 2026-03-31
>
> Dear Reviewer A2Vs,
>
> Thanks for your insightful reviews. Below, we provide point-by-point responses to each of your comments.
>
> > W1: The method is a thoughtful combination of known ideas
>
> We thank the reviewer for this thoughtful assessment. We agree that QVG does not claim a fundamentally new primitive in isolation; rather, our contribution is to formulate video KV-cache compression as a distinct systems problem for long-horizon auto-regressive video generation, and to show that existing LLM-style KV quantization methods fail in this setting due to the combination of strong spatiotemporal redundancy and severe token/channel heterogeneity. QVG is designed specifically around this structure, through semantic-aware smoothing, progressive residual refinement, and a streaming-oriented implementation that together make ultra-low-bit KV-cache compression practical for autoregressive video models for the first time, enabling up to 7× compression with modest overhead while preserving long-horizon generation quality.
>
> > W2 & Q1: Sensitivity and robustness are not fully characterized
>
> Our main conclusion is that
> 1. **QVG remains stable** under different number of clusters, chunking strategies, numbers of progressive stages, and initialization choices.
> 2. We recommend the following setting **across models and resolutions**: number of centroids K = 256, precision = INT2, chunk size $\approx$ 40k, number of PRQ stages = 1 or 4, quantization group size = 64.
>
> Due to space limitations, we summarize the high-level takeaways here and refer to the detailed results and tables in our other responses.
>
> For progressive stages, the paper already includes an ablation on the number of progressive stages in Figure 5c and Section 5.4, which shows stable behavior and a clear quality-memory trade-off.
>
> For cluster numbers, chunking strategy and initialization, we add ablations to show that the method is not sensitive to these choices. Details in responses to Reviewer tk8y (Q4, Q3-1, and Q3-2, respectively)
>
> For transferability across resolutions, we observe that the same default configuration continues to perform well. Details in response to Reviewer dKUT (Q3).
>
>
> > W2 & Q2: Highly dynamic scenes, rapid shot changes
>
> We further evaluate QVG on challenging cases with rapid motion, scene changes, and low redundancy. QVG still clearly outperforms prior baselines, validating its effectiveness.
>
> |Method|Compression Ratio|PSNR|SSIM|LPIPS|
> |-|-|-|-|-|
> |KIVI|$6.40\times$|19.836|0.698|0.235|
> |Quarot|$6.40\times$|20.733|0.724|0.195|
> |QVG|$6.94\times$|27.941|0.879|0.083|
>
> > W3 & Q3: Overhead and scaling concerns
>
> The overhead of QVG is mainly determined by chunk size, rather than directly by resolution, sequence length, or batch size. Because QVG operates on fixed-size KV-cache chunks, increasing resolution or sequence length mainly increases the number of chunks, while the per-chunk cost stays unchanged, keeping the relative overhead low and stable.
>
> Similarly, as batch size increases, both the model forward and QVG bookkeeping scale roughly linearly, while the model forward remains dominant. As a result, larger batch sizes increase absolute latency but have little effect on QVG’s percentage overhead. We conduct an additional experiment to validate this.
>
> |Batch size|End-to-end|QVG extra cost|Total overhead|
> |-|-|-|-|
> |1|43 s|0.74 s|1.7%|
> |2|86 s|1.4 s|1.6%|
> |5|217 s|3.7 s|1.7%|
>
> > W4 & Q4: Breadth of evaluation
>
> Answer: Our current evaluation is focused on the generation pipeline of autoregressive video generation with a growing KV cache. Within this scope, we **already study multiple model families** (LongCat Video, Self-Forcing, Hunyuan-WorldPlay, each with a different tokenizer) and decoding patterns, including fixed-window continuation and full-history chunk-wise generation. Therefore, we believe the current coverage is **already well aligned with the paper's target problem setting** and can **support the model-agnostic claim**.
>
>
> > W5: Reproducibility details
>
> These mentioned details are **provided in Section 5.1**, including the GPU platform (NVIDIA H100, CUDA 12.8), kernel implementation (custom CUDA + Triton), precision / quantization settings (e.g., FP8 E4M3 scaling, INT2/INT4 configurations), caching policy (streaming chunk-wise compression and pre-RoPE key caching), and the exact evaluation metrics / baseline setup.
>
> > Q5: Which part contributes most to long-horizon stability?
>
> Ablations in Section 5.4 suggest that long-horizon stability mainly comes from reducing quantization error, with **semantic-aware smoothing (SAS) being the dominant contributor**. As shown in Figure 5c, the first stage **reduces the error dramatically** (by 5.83$\times$), while additional stages in PRQ further reduce the error by 1.09$\times$ ~ 1.29$\times$.
>
> By contrast, the fused reconstruction kernel is only a systems optimization: it speeds up dequantization but does not change the reconstructed values, and thus does not affect long-horizon quality.

---

> > ### Author Rebuttal · Reviewer_A2Vs · 2026-04-01
> >
> > My concerns have been adequately addressed.

---

### Official Review · Reviewer_ekUk · 2026-02-25

**Soundness:** 3
**Presentation:** 3
**Significance:** 3
**Originality:** 3
**Overall Recommendation:** 4
**Confidence:** 2

**Summary:**

The authors reveal that KV-cache memory is a system and algorithm bottleneck affecting long-horizon generation quality and deployability in autoregressive video diffusion. To solve this bottleneck, they present Quant VideoGen (QVG), a training-free KV-cache quantization framework tailored to video models. QVG introduces Semantic-Aware Smoothing to exploit spatiotemporal redundancy and produce low-magnitude, quantization-friendly residuals, followed by Progressive Residual Quantization to iteratively reduce the quantization error. Evaluations on LongCat-Video, HY-WorldPlay, and Self-Forcing show up to 7× KV memory reduction with minimal latency overhead and near-lossless visual quality, enabling deployment on more modest hardware.

**Compliance With Llm Reviewing Policy:**

Affirmed.

**Final Justification:**

All my concerns are solved, so I would like to keep my rating.

**Key Questions For Authors:**

I am not an expert in AR Quantization, and not familiar with the related SOTA. Back to the proposed method, the solution is reasonable and the performance is great.

**Strengths And Weaknesses:**

Strengths:
1. Tackles a pressing limitation in autoregressive video generation by explicitly linking KV-cache size to long-horizon consistency, and proposing a practical system-level solution.
2. Comprehensive evaluation across multiple models, dataset benchmarks, precisions (INT2/INT4), and metrics (PSNR, SSIM, LPIPS, VBench) demonstrates consistent improvements over strong baselines (RTN, KIVI, QuaRot).
3. The entire framework is training-free, making it broadly applicable to existing deployed models.

Weaknesses:
1. The evaluation focuses on a limited set of autoregressive video models; broader testing on varied architectures and tasks (e.g., bidirectional or hybrid models) could strengthen generality claims.
2. While comparisons with LLM-oriented quantization baselines are included, there is less exploration of alternative video-specific compression strategies beyond quantization.
3. The choice of k-means group counts and centroid storage accuracy are not deeply analyzed in terms of their effect on quality and compression ratio.

---

> ### Author Rebuttal · Authors · 2026-03-31
>
> Dear Reviewer ekUk,
>
> Thank you for your valuable questions. Below, we address each point raised.
>
> > W1: Broader testing on varied architectures and tasks (e.g., bidirectional or hybrid models).
>
> Answer: QVG is **designed for auto-regressive video generation with a growing KV cache**. By contrast, bidirectional full-attention video models do not have KV cache. Linear-attention video models (e.g., SANA-Video) use a constant size KV cache that do not grow over time. For hybrid full-attention/linear-attention video architectures, to the best of our knowledge, there is not yet a widely adopted open-source models. Therefore, we believe the current coverage is already well aligned with the paper’s target problem setting.
>
> > W2: Comparison with alternative video-specific compression strategies beyond quantization.
>
> We thank the reviewer for this important suggestion. We add two practical history-dropping baselines: (1) the native Self-Forcing sliding-window policy, and (2) a sink+window policy adapted from SDV2 (StreamDiffusionV2, MLSys 2026). We compare them with QVG under a matched KV-memory budget and report PSNR / SSIM / LPIPS against the BF16 reference. The results show that **QVG achieves a better quality-memory trade-off**.
>
>
>
> |           Method           | PSNR | SSIM | LPIPS |
> |:--------------------------:|:----:|:----:|:-----:|
> |   Native Sliding Window    |   16.61   |   0.5094   |   0.3131    |
> | Sink + Window (SDV2-style) |   21.48   |   0.7426   |    0.1471   |
> |            QVG             |   27.42   |   0.8729   |   0.0769    |
>
>
>
> > W3: The choice of k-means group counts and centroid storage accuracy are not deeply analyzed in terms of their effect on quality and compression ratio.
>
> Answer:
> We thank the reviewer for the comment. The choice of K should balance clustering quality and metadata overhead. Increasing K improves clustering quality, but also increases index/codebook overhead and reduces the compression ratio.
>
> In the table below, we sweep over the codebook size K ∈ {64, 128, 256, 512} while fixing the number of stages to 1, and report results under both INT2 and INT4 settings. We report the resulting compression ratio together with reconstruction improvement for the key and value caches. Here, K-Imp and V-Imp denote the improvement of MSE over naive quantization, so larger values indicate better reconstruction quality and lower error.
>
> We find that both K = 128 and K = 256 **provide good tradeoffs**. We choose K = 256 in the final design because it allows the assignment vector to be **stored naturally in uint8**, i.e., each token’s cluster ID fits in one byte.
>
> INT4
>
> | Codebook K | Compression ratio | K-Imp | V-Imp |
> |:-:|:-:|:--:|:-:|
> |512|3.819| 6.558 | 2.391 |
> |256|3.881| 5.944 | 2.229 |
> |128|3.917| 5.402 | 2.083 |
> |64|3.939| 4.912 | 1.946 |
>
> INT2
>
> | Codebook K | Compression ratio | K-Imp | V-Imp |
> |:-:|:-:|:-:|:-:|
> |512|7.307| 6.428 | 2.356 |
> |256|7.539| 5.834 | 2.199 |
> |128|7.676| 5.307 | 2.056 |
> |64|7.760| 4.832 | 1.924 |

---

> > ### Author Rebuttal · Reviewer_ekUk · 2026-04-05
> >
> > Thanks for the rebuttal, and all my concerns are solved.

---

> > > ### Author Response · Authors · 2026-04-06
> > >
> > > Reviewer ekUk,
> > >
> > > We really appreciate your insightful and constructive comments, which helped us improve this paper. If you feel that your concerns have been resolved, we would greatly appreciate it if you consider raising the score.
> > >
> > > Best wishes!
> > >
> > > The Authors

---

### Official Review · Reviewer_dKUT · 2026-03-08

**Soundness:** 3
**Presentation:** 3
**Significance:** 3
**Originality:** 3
**Overall Recommendation:** 4
**Confidence:** 5

**Summary:**

This paper introduces Quant VideoGen (QVG), a training-free 2-bit KV-cache quantization framework designed specifically for auto-regressive video diffusion models. The authors identify that video KV-cache suffers from severe memory bottlenecks (e.g., ~34 GB for 5 seconds of 480p video) , and that directly applying LLM-centric quantization methods fails due to highly dynamic numeric ranges and token heterogeneity. To solve this, QVG leverages spatiotemporal redundancy via Semantic-Aware Smoothing (using k-means to group similar tokens and subtract their centroids). It then iteratively applies this via Progressive Residual Quantization to compress the cache. Extensive evaluations show QVG can reduce KV-cache by up to 7.0x with near-lossless quality and minimal latency overhead.

**Compliance With Llm Reviewing Policy:**

Affirmed.

**Final Justification:**

The rebuttal solved my concerns.

**Key Questions For Authors:**

- Algorithmic Distinction: Could you elaborate on how Progressive Residual Quantization algorithmically differs from standard Residual Vector Quantization (RVQ) used in audio/video codecs, aside from the fact that it is applied to the KV-cache?

- Latency Jitter: In Section 4.3, you mention centroid caching reduces k-means overhead. Does the number of required k-means iterations fluctuate significantly across different video chunks? How does this impact the maximum tail latency (e.g., P99 latency) during streaming generation?

- Scalability to Higher Resolutions: The experiments are conducted at 480p. As resolution scales to 720p or 1080p, the number of tokens per frame increases quadratically. Does the optimal number of groups ($C$) or centroids need to scale linearly with this increase to maintain the same quantization error bounds?

**Limitations:**

No. This paper requires a discussion section.

**Strengths And Weaknesses:**

Strengths:
- Highly Relevant and Impactful Problem: Addressing the KV-cache bottleneck in auto-regressive video models is a critical system challenge. By reducing the memory footprint by 7x , this work practically enables long-horizon generation on consumer-grade hardware (e.g., fitting HY-WorldPlay-8B on an RTX 4090).

- Video-Specific Algorithmic Design: The authors astutely observe that video tokens exhibit different numeric distributions than text , but also contain massive spatiotemporal redundancy. Semantic-Aware Smoothing is a clever, well-motivated solution to regularize these distributions before quantization.

- Strong Empirical Results: QVG decisively outperforms existing state-of-the-art LLM KV-cache quantizers like KIVI and QuaRot, maintaining excellent PSNR (>28.7) and VBench scores even at extreme compression ratios.

 Weaknesses:
- Clarification on Algorithmic Novelty of PRQ: The Progressive Residual Quantization (PRQ) mechanism essentially applies iterative residual quantization. While the application to video KV-cache is highly effective, the fundamental mathematical concept is a well-established technique in traditional streaming codecs and neural compression (e.g., Residual Vector Quantization). The paper could better clarify the boundary between existing residual quantization literature and their specific architectural adaptations.

- Overhead and Stability of On-the-Fly Clustering: The method relies on k-means clustering during the forward pass. While the authors introduce a "streaming centroid caching" trick to reduce overhead, k-means can still introduce latency jitter depending on convergence rates. A deeper analysis of worst-case latency spikes during real-time streaming would strengthen the system-level evaluation.

---

> ### Author Rebuttal · Authors · 2026-03-31
>
> Dear Reviewer dKUT,
>
> We thank the reviewer for the acknowledgment of our novel contributions. Below we respond to the questions.
>
> > W1 & Q1: Clarification on Algorithmic Novelty of PRQ.
>
> Answer: We thank the reviewer for the comment. As stated in our paper, our paper explicitly notes that PRQ is inspired by streaming video codecs. We agree that PRQ is not intended to be a fundamentally new residual quantization principle in isolation. Rather, the novelty lies in the **video-KV-specific formulation, integration, and deployment recipe**, which together lead to **strong empirical gains**. Our contribution is adapting residual refinement compression to the challenging setting of online video KV-cache quantization, together with the semantic grouping mechanism and the system co-design that make it effective in practice. Empirically, this combination enables up to **7× KV compression with modest latency overhead** while preserving near-lossless quality.
>
> Specifically, our method is built around three components working together:
> (1) Semantic-Aware Smoothing, which groups semantically similar KV tokens and subtracts centroids to make the residuals much more quantization-friendly
> (2) PRQ, which applies multi-stage residual refinement in a training-free way on top of this transformed representation
> (3) streaming-oriented system design, including efficient centroid reuse and fused dequantization, to make the method practical during autoregressive video generation.
>
> We will make this clearer in the camera ready version by explicitly distinguishing existing traditional streaming codecs from our main contribution: a video KV Cache specific compression framework that combines semantic smoothing, progressive refinement, and efficient deployment into a single practical system.
>
> > W2 & Q2: Overhead and Stability of On-the-Fly Clustering.
>
> We thank the reviewer for this question. We do not rely on full k-means convergence in practice, but instead **run a fixed number of iterations**. This design stabilizes latency and eliminates iteration-dependent jitter across chunks. In our implementation, the k-means routine has a fixed maximum iteration cap of 20, and after enabling streaming centroid caching, the clustering converges much faster: 5–6 iterations already achieve essentially the same PSNR as 20 iterations. We add a small ablation showing that centroid caching substantially reduces the k-means overhead while preserving quality.
>
> |Centroid Cache|# Iter|PSNR|K-means overhead|
> |---|---|---|---|
> |$\times$|6|26.94|1.7%|
> |$\times$|20|27.39|4.9%|
> |$\checkmark$|6|27.42|1.7%|
>
> > Q3: Scalability to Higher Resolutions
>
> We thank the reviewer for this question. To assess the scalability beyond 480p, we evaluate QVG on LongCat-Video at 720p and keep the same centroid configuration as in 480p. We observe that QVG consistently outperforms the baseline quantization methods and maintain the same error bounds.
>
> |Method|Compression Ratio|PSNR|SSIM|LPIPS|
> |---|---|---|---|---|
> |KIVI|$6.4\times$|17.66|0.5518|0.3441|
> |Quarot|$6.4\times$|20.88|0.6659|0.2395|
> |QVG|$6.9\times$|25.99|0.8174|0.1177|
>
> Notably, we do not need to increase the number of centroids when moving from 480p to 720p. The reason is that QVG performs quantization on fixed-size KV-cache chunks, rather than on all tokens of a frame. When the spatial resolution increases, each frame contributes more tokens, so the number of frames contained in each chunk decreases accordingly, while the per-chunk clustering scale remains unchanged.

---

> > ### Author Rebuttal · Reviewer_dKUT · 2026-04-02
> >
> > My concerns have been resolved and this paper can be accepted after adding the new experiments to the main paper.

---

> > > ### Author Response · Authors · 2026-04-05
> > >
> > > Dear Reviewer dKUT,
> > >
> > > We really appreciate your insightful and constructive comments, which helped us improve this paper. If you feel that your concerns have been resolved, we would greatly appreciate it if you consider raising the score.
> > >
> > > Best wishes!
> > >
> > > The Authors

---

### Official Review · Reviewer_tk8y · 2026-03-12

**Soundness:** 2
**Presentation:** 3
**Significance:** 2
**Originality:** 3
**Overall Recommendation:** 3
**Confidence:** 5

**Summary:**

This paper presents Quant VideoGen, a training-free KV-cache quantization framework for auto-regressive video diffusion models.
Since the KV cache may grow and dominate GPU memory during generation.
The paper introduces two techniques:1) Semantic-Aware Smoothing, which clusters tokens via k-means based on latent similarity and subtracts group centroids to produce low-magnitude quantization friendly residuals. 2) Progressive Residual Quantization, which follows a corase-to-fune multi-stage scheme that further reduces quantization error while enabling a smooth quality memory trade-off.
The evaluations show better quality compared to baseline methods.

**Compliance With Llm Reviewing Policy:**

Affirmed.

**Final Justification:**

The paper features sufficient experiments and addresses a valid deployment bottleneck, but given the limited theoretical innovation, my rating remains borderline, though I would not object to it being accepted.

**Key Questions For Authors:**

1. The paper claims minute-level generation. The longest experiments seem to be ~700 frames. It would be more convincing to generate longer videos.
2. The paper only compared with the quantization baselines, while the more common and practical approach on KV cache compression in practice is sink-window attention mechanisms, which are not in the evaluation.
3. The cost of k-means is underexplored under different block/chunk sizes. A profiling for the overhead of k-means under different block/chunk sizes can be helpful to better understand the trade-off. The k-means is also non-deterministic, different random seeds can yield different clusterings. The paper didn't discuss this.
4. Why choose on K=256? Is there an analysis for it?

**Limitations:**

yes

**Strengths And Weaknesses:**

1. The paper is well motivated, since the KV cache is a deployment bottleneck for auto-regressive generation.
2. The paper explores the spatio-temporal redundancy in video tokens. The cosine similarity analysis provides good motivation.
3. The improvement outperforms baselines at Int2 by a large margin.
4. The end-to-end overhead is under 4%, and it is a notable practical strength. But it is only measured on the H100 GPU. Overhead on consumer hardware could be more helpful.

---

> ### Author Rebuttal · Authors · 2026-03-31
>
> Dear Reviewer tk8y,
>
> Thanks for your insightful reviews. Below, we provide point-by-point responses to each of your comments.
>
> > W1: Overhead on consumer hardware.
>
> We add a breakdown on Self-Forcing for both RTX 5090 and H100, reporting the end-to-end latency and the quantization-related overhead. The new results show that QVG remains lightweight on both datacenter and consumer-class GPUs.
>
> | Platform | End-to-end | QVG extra cost | Total overhead |
> |-|-|-|-|
> | H100 | 43 s | 0.74 s | 1.7% |
> | RTX 5090 | 72 s | 1.34 s | 1.9% |
>
> We observe that the quantization overhead **remains very small** on both platforms, staying below 2% of the end-to-end latency. The overhead is slightly higher on RTX 5090 than on H100, which is consistent with the slower consumer-class GPU making the quantization cost a slightly larger fraction of the total runtime. Overall, these results show that QVG introduces only **marginal extra cost across different hardware settings**.
>
>
> > Q1: Generate and test on longer videos
>
> We further extend the Self-Forcing evaluation to **1400 frames (about 90s)** under the same generation setup as in the paper, and report the VBench Image Quality of BF16, KIVI, QuaRot, and QVG under INT2 quantization setting. QVG consistently **achieves image quality similar to BF16** as the video length increases, while the other baselines degrade much more noticeably. These results further support the effectiveness of QVG.
>
> | Method | 350 frames | 700 frames | 1050 frames | 1400 frames |
> |-|-|-|-|-|
> |  BF16  |   0.7433   |   0.7056   |   0.6880    |   0.6587    |
> |  KIVI  |   0.6757   |   0.5718   |   0.4431    |   0.3585    |
> | QuaRot |   0.4833   |   0.4517   |   0.4558    |   0.4445    |
> |  QVG   |   0.7436   |   0.6952   |   0.6723    |   0.6728    |
>
>
> > Q2: Comparison with sink-window attention mechanisms
>
> We add two practical history-dropping baselines: (1) the native Self-Forcing sliding-window policy, and (2) a sink+window policy adapted from SDV2 (StreamDiffusionV2, MLSys 2026). We compare them with QVG under a matched KV-memory budget and report PSNR / SSIM / LPIPS against the BF16 reference. The results show that QVG achieves a **better quality-memory trade-off**.
>
> | Method | PSNR | SSIM | LPIPS |
> |-|-|-|-|
> | Native Sliding Window | 16.61 | 0.5094 | 0.3131 |
> | Sink + Window (SDV2-style) | 21.48 | 0.7426 | 0.1471 |
> | QVG | 27.42 | 0.8729 | 0.0769 |
> > Q3 - 1: Overhead of k-means under different block/chunk sizes
>
> In the table below, we report the profiling results on Self-Forcing model under the INT2 setting while varying the chunk size.
>
> | Chunk Size | Frames per Chunk | Per-Layer KV Cache Memory | Compression Ratio | Overhead |
> |-|-|-|-|-|
> | 37440 | 24 | 220MB | 7.0x | 1.3% |
> | 18720 | 12 | 230MB | 6.7x | 2.0% |
> | 9360  | 6  | 264MB | 5.8x | 3.3% |
>
> These results show that the k-means / quantization **overhead remains small across a wide range of chunk sizes** (maximum of 3.3% of the overall run time). Larger chunk sizes are clearly more favorable for memory efficiency, yielding both lower overhead and higher compression ratio. This is because clustering and metadata costs are better amortized over more tokens within each chunk. Overall, the results suggest that **QVG is practical across different chunk sizes**, while larger chunks leads to better compression ratio.
>
> > Q3 - 2: Non-determinism of k-means
>
> We evaluated the sensitivity to k-means initialization on Self-Forcing using different random seeds under both INT4 and INT2 settings.
>
> | Precision | seed 0 | seed 1 | seed 2 | Max diff |
> |-|-|-|-|-|
> | INT4 | 0.7453 | 0.7351 | 0.7347 | 0.0106 |
> | INT2 | 0.7461 | 0.7348 | 0.7345 | 0.0116 |
>
> The variation is very small, indicating that **QVG is robust to the non-determinism** of k-means initialization in practice.
>
> > Q4: Why choose on K=256?
>
> Answer:
> The choice of K should balance clustering quality and metadata overhead. In the table below, we sweep over the codebook size K ∈ {64, 128, 256, 512} while fixing the number of stages to 1, and report results under both INT2 and INT4 settings. We report the resulting compression ratio together with reconstruction improvement for the key and value caches. Here, K-Imp and V-Imp denote the improvement of MSE over naive quantization, so larger values indicate better reconstruction quality and lower error.
>
> We find that both K = 128 and K = 256 **provide good tradeoffs**. We choose K = 256 in the final design because it allows the assignment vector to be **stored naturally in uint8**, i.e., each token’s cluster ID fits in one byte.
>
> INT4
>
> | Codebook K | Compression ratio | K-Imp | V-Imp |
> |--|--|--|--|
> | 512 | 3.819 | 6.558 | 2.391 |
> | 256 | 3.881 | 5.944 | 2.229 |
> | 128 | 3.917 | 5.402 | 2.083 |
> | 64 | 3.939 | 4.912 | 1.946 |
>
> INT2
>
> | Codebook K | Compression ratio | K-Imp | V-Imp |
> |--|--|--|--|
> | 512 | 7.307 | 6.428 | 2.356 |
> | 256 | 7.539 | 5.834 | 2.199 |
> | 128 | 7.676 | 5.307 | 2.056 |
> | 64 | 7.760 | 4.832 | 1.924 |

---

> > ### Author Rebuttal · Reviewer_tk8y · 2026-04-04
> >
> > Thanks for the author's rebuttal.

---

> > > ### Author Response · Authors · 2026-04-05
> > >
> > > Dear Reviewer tk8y,
> > >
> > > We really appreciate your insightful and constructive comments, which helped us improve this paper. If you feel that your concerns have been resolved, we would greatly appreciate it if you consider raising the score.
> > >
> > > Best wishes!
> > >
> > > The Authors

---

### Decision · Program_Chairs · 2026-04-30

**Decision:**

Accept (regular)

**Comment:**

This paper studies KV-cache quantization for autoregressive video generation. It adapts techniques that are well-established in large language models to the video domain, addressing a practical and timely problem. All reviewers acknowledged the significant empirical gains achieved by the proposed plug-and-play method.

Reviewers raised several concerns, including quantization and clustering overhead, generalization to other architectures and longer video generation, the need for more ablations on hyperparameters (e.g., the choice of K), and the technical novelty of the proposed Progressive Residual Quantization relative to existing quantization approaches. The rebuttal addressed most of these concerns with additional experiments and clarifications.

Overall, all reviewers are either in favor of acceptance or not opposed to it. The AC concurs and recommends acceptance. The authors are encouraged to incorporate the additional results from the rebuttal into the final version.